# Automatic Modulation Classification Using Hybrid Data Augmentation and Lightweight Neural Network

**DOI:** 10.3390/s23094187

**Published:** 2023-04-22

**Authors:** Fan Wang, Tao Shang, Chenhan Hu, Qing Liu

**Affiliations:** 1National Key Laboratory of Integrated Service Networks, Xidian University, Xi’an 710071, China; wangf@crirp.ac.cn; 2China Research Institute of Radiowave Propagation, Qingdao 266107, China; 3Glasgow College, University of Electronic Science and Technology of China, Chengdu 611731, China

**Keywords:** automatic modulation classification (AMC), lightweight neural network, data augmentation, deep learning

## Abstract

Automatic modulation classification (AMC) plays an important role in intelligent wireless communications. With the rapid development of deep learning in recent years, neural network-based automatic modulation classification methods have become increasingly mature. However, the high complexity and large number of parameters of neural networks make them difficult to deploy in scenarios and receiver devices with strict requirements for low latency and storage. Therefore, this paper proposes a lightweight neural network-based AMC framework. To improve classification performance, the framework combines complex convolution with residual networks. To achieve a lightweight design, depthwise separable convolution is used. To compensate for any performance loss resulting from a lightweight design, a hybrid data augmentation scheme is proposed. The simulation results demonstrate that the lightweight AMC framework reduces the number of parameters by approximately 83.34% and the FLOPs by approximately 83.77%, without a degradation in performance.

## 1. Introduction

Adapting the modulation used for changing channel conditions is a crucial way to maximize the throughput of a communication link and maintain continuous link reliability in a mobile environment [1,2,3,4]. By changing the modulation method, the transmitted signal can adjust to the changing channel environment, minimizing the difference between channel capacity and throughput without sacrificing robustness [5]. Automatic modulation classification (AMC) [6,7,8,9,10,11] is a crucial technology for non-cooperative communications, serving as an intermediate step in signal detection and demodulation in communications. AMC has a wide range of applications in both military and civilian domains. In the military field, AMC can interfere with the enemy modulation signal when identified, while in the civilian field, it is the basis for spectrum allocation and signal demodulation [12]. Given the current increasingly complex electromagnetic environment, AMC technology has garnered significant attention from researchers.

The general AMC process comprises two primary steps: signal preprocessing and classification. Signal preprocessing involves noise reduction and estimation of signal parameters, such as carrier frequency and signal power. The classification step in AMC methods can be broadly categorized into likelihood ratio-based methods [13,14] and feature-based methods [15,16,17]. The likelihood ratio-based method [18] uses the modulated signal to establish the likelihood function, and then compares the calculated likelihood ratio function value with the threshold value to complete the signal classification. The likelihood ratio-based AMC method yields the optimal solution at the cost of high computational effort. It has high computational complexity and requires channel state information of the communication system. The feature-based AMC method has relatively low computational complexity but relies heavily on specialized knowledge to construct complex feature engineering. This includes higher-order statistics-based, parametric statistics [19], constellation diagrams, cyclic spectrum [20], time-frequency transformation domain, and others. Moreover, the classifier is designed for specific features, such as artificial neural networks (ANNs), support vector machines (SVMs), K-nearest neighbors (KNNs), etc.

In recent years, deep learning techniques have made significant breakthroughs in the fields of image [21] and natural language processing [22]. Hence, many researchers have attempted to apply deep learning techniques to AMC. Artificial intelligence has numerous applications in wireless communication, including signal detection [23,24,25], channel estimation [26,27,28,29,30], radio frequency fingerprint identification [31,32,33], and beamforming [34,35]. Prior to the development of artificial intelligence, classical methods mainly designed features through signal processing technologies, and then machine learning-based classifiers were applied for identification. However, there are many shortcomings of the characteristics designed by artificial knowledge and experience, such as weak performance, poor environmental adaptability, and difficulty in dealing with increasingly complex and numerous wireless transmitters. From a processing flow perspective, AMC using deep learning methods is primarily based on image transformation and original sampled signals. Image transformation-based AMC involves converting the original sampled signal into an image and then using deep learning to classify modulations, such as images. The commonly used image transformations include time-frequency diagrams [36,37,38], constellation maps [39], and others. On the other hand, deep learning is used to perform AMC on the sampled signal [40]. The sampled signal already contains the necessary information for AMC, without increasing the computational effort of conversion to an image, making it suitable for application in spatially embedded computers.

Convolutional neural networks (CNNs) are the most widely used network structures in image processing and long short-term memory (LSTM) networks [41,42] are widely used in natural language processing (NLP); both are gradually receiving attention from experts in the field of automatic modulation classification. O’shea et al. [43] first introduced CNN into the field of automatic modulation classification and proposed an AMC method based on CNN. West et al. [44] introduced a network structure combining CNN and LSTM, such as CLDNN [45], with an accuracy of more than 80% for the classification of 11 modulations when the signal-to-noise ratio (SNR) of ≥0 dB. O’shea et al. [46] transformed the two-dimensional convolution in VGG [47] and ResNet [48]; Lin et al. [49] used one-dimensional convolution for extracting time-domain features with an accuracy of over 90% for the classification of 24 modulations when SNR ≥ 10 dB. Krzyston et al. [50] fully considered the characteristics of electromagnetic signals, treated the in-phase quadrature (IQ) signals as complex signals, and designed a convolutional neural network in the complex form; its accuracy exceeded 80% when SNR ≥ 10 dB in the classification of 11 schemes.

Unfortunately, however, the above methods only compare the classification accuracy and do not investigate the parameters, computational efforts, and complexity. Due to the limitations of size and power consumption, the storage space and computational power of lightweight equipment are significantly restricted. Furthermore, deep neural networks have a large number of parameters and high computational complexity, limiting their application in communication systems. To address this challenge, reduce the size of the network model, decrease its complexity, and increase the model inference speed, researchers in the field of image processing [51] have proposed lightweight network structures, such as SqueezeNet [52], MobileNet [53], and ShuffleNet [54] by optimizing the structure of neural networks.

In this paper, we propose a lightweight complex-valued residual network (CVResNet)-based AMC method. A hybrid data augmentation method was designed to compensate for the decrease in classification performance resulting from the lightweight design. The main contributions of this paper are summarized as follows.

A lightweight residual network based on complex-valued operations is proposed, named CVResNet, aiming to address the high computational effort and complexity of traditional non-lightweight networks.A hybrid data augmentation method is designed to compensate for the potential performance degradation caused by the lightweight network.Comparative experiments are conducted in the same scenario, and the results demonstrate that CVResNet can significantly reduce computational complexity and effort while achieving better classification performance.

## 2. Signal Model and Problem Formulation

### 2.1. Signal Model

The signal data in this paper take into account the effects of the phase shift and additive Gaussian noise; the signal model can be expressed as
(1)$$\begin{array}{c} x\left(n\right)=he^{j\theta (n)}s\left(n\right)+w\left(n\right),n\in 0,1,2,\cdots ,N-1, \end{array}$$
where s(n) denotes the transmitted baseband signal sequence, *h* denotes the channel gain, θ(n) denotes the time-varying phase offset, and w(n) represents the additive white Gaussian noise. *N* is the number of sampling points of the signal. The radio signal x(n) received by the receiver consists of in-phase (I) and quadrature (Q) components, which can be seen as the real and imaginary parts of x(n), respectively. x(n) can be expressed as
(2)$$\begin{array}{c} x\left(n\right)=\left[\begin{array}{c} I\\ Q \end{array}\right]=\left[\begin{array}{c} real(x(n))\\ imag(x(n)) \end{array}\right]. \end{array}$$

### 2.2. System Model

The framework of the deep learning-based AMC system is shown in Figure 1. In this system, the classifier is critical, as it needs to classify the modulation scheme based on the pre-processed data.

In the training process, the goal is to obtain a mapping function f∈F:X→Y with a low number of parameters. X and Y represent the sample space and category space, respectively. Due to the limited amount of data, the neural network can approximate the mapping function between X and Y by finding the mapping relationship between existing data and labels in the database. The optimization target can be expressed as
(3)$$\begin{array}{c} \min_{f\in F}E_{(x,y)∼D}\left\{L_{ce}\left[f\left(x\right),y\right]\right\}, \end{array}$$
where *D* is the existing dataset, *y* is the modulation category label of *x*, and $L_{ce}$ is the cross-entropy loss.

In the testing process, the neural network is employed as the classifier. For a communication system, an accurate and fast classifier facilitates the subsequent correct demodulation and the stability of the whole system. The classifier infers the probability that the input signal belongs to each modulation type in the modulation type pool $P=\{p_{j},j=1,2,\cdots ,J\}$, where *J* represents the number of modulated types and $p_{j}$ represents the modulated type. The inference of the classifier is shown as
(4)$$\begin{array}{c} p=\underset{p_{j}\in P}{\arg\max}P\left(p_{j}|x\right), \end{array}$$
where $P(p_{j}|x)$ is the probability that the input signal x belongs to modulation type $p_{j}$ and *p* is the classification result.

## 3. Our Proposed AMC Method

In this section, the proposed AMC method is described in four subsections, including the framework of the proposed AMC method, details of the lightweight CVResNet, hybrid data augmentation, and the training procedure.

### 3.1. Framework of the Proposed AMC Method

The framework of the proposed AMC method is shown in Figure 2. Our proposed AMC method is composed of a training process and a testing process. In the training process, the parameters of an excellent lightweight network are trained using a hybrid DA method. In the testing process, the lightweight network is used for modulating signal classification.

### 3.2. Details of the Lightweight CVResNet

#### 3.2.1. Deep Residual Neural Network

In neural network models, the depth of the network has a significant impact on the performance of the model. Deeper networks can perform more complex and powerful feature extraction, but this also leads to problems such as gradient disappearance, gradient explosion, and network degradation. To avoid these problems, this paper chooses the residual network as the feature extraction network for AMC. The residual network and the residual unit (Resunit) are shown in Figure 3. In the residual network, the Resunit extracts the features, the average pooling (AvgPool) reduces the feature dimension, and ’linear’ is the fully connected layer that classifies the features to obtain modulated classification results.

#### 3.2.2. Complex Convolution

Regular neural networks are defined in the real-valued domain and their weight parameters and the data passed in the network are real-valued numbers. The complex-valued neural network can be considered an extension of the conventional neural network from the real-valued domain to the complex-valued domain, where the network parameters are all complex-valued numbers and allow the input of the network to be in the form of complex-valued numbers. Therefore, it can be used to process modulated signals in IQ format and better retain the information contained in the modulated signals.

To reduce the complexity caused by complex-valued neural networks, only complex convolution [55] is used in this paper. As shown in Figure 4, the idea is to treat the real and imaginary parts of the complex numbers as logically distinct real-valued subjects and to simulate complex operations internally with real-valued arithmetic. By defining the input complex-valued data S=I+iQ and the convolution kernel K=a+ib, where a and b are independent real convolution kernels, the convolution result of S with K can be expressed as
(5)$$\begin{array}{c} S*K=\left(I*a-Q*b\right)+i\left(Q*a+I*b\right), \end{array}$$
where ∗ is the convolution operation. From this formula, it can be seen that complex convolution is the conversion of the convolution of complex numbers with complex numbers into the convolution of real numbers with real numbers, and combining them, so that complex convolution can be achieved without modifying the real-valued convolution operation.

#### 3.2.3. Depthwise Separable Convolution

Despite the powerful feature extraction capabilities of complex-convolutional residual networks, the complex structure, large number of parameters, and long training time make it difficult to deploy the complex, convolutional residual networks on actual lightweight devices for AMC. To solve the problem, this paper combines complex convolution with depthwise separable convolution and proposes a lightweight depthwise separable complex convolution. It is done by replacing the real convolution in the complex convolution with a depthwise separable convolution [56,57]; this method can significantly reduce the number of parameters of convolutional layers. As shown in Figure 5, the depthwise separable convolution consists of depthwise convolution and pointwise convolution.

Each convolution kernel is responsible for one channel, and each channel is convolved by only one convolution kernel. This process generates the same number of channels in the feature map as the number of channels in the input. The operation of pointwise convolution is similar to the regular convolution operation, with a convolution kernel of size 1×1×W, where *W* is the number of channels in the previous layer. Here, the convolution operation combines the maps from the previous layer in the depth direction with weighting to generate a new feature map.

Although depthwise separable convolution reduces computational complexity and parameter count, it can provide similar performance to ordinary convolutions in practical applications. This is mainly because—in many automatic modulation recognition tasks—the correlation between spatial features and channel features is relatively weak. By separating these two types of features, depthwise separable convolution can significantly reduce computational costs with minimal performance loss.

### 3.3. Hybrid Data Augmentation

Data augmentation methods can prevent models from overfitting, improve model robustness, and address sample imbalances. To compensate for the potential performance degradation caused by using lightweight networks, we propose to improve classification performance by using a hybrid data augmentation method. The flow of this hybrid data augmentation method is shown in Figure 6. Specifically, the hybrid data augmentation method is implemented using both rotation, which expands the number of samples in the dataset, and RandMix, which serves as a regularization method to enhance model robustness and improve generalization performance. The hybrid data augmentation method is described as follows.

#### 3.3.1. Rotation

The rotation data augmentation method works directly on the original signal dataset and serves to expand the number of samples in the dataset to improve the generalization ability of the model. The principle of rotation data augmentation is shown in Figure 7. Specifically, for a given original signal sample (I,Q), the augmented samples $\left(I^{\prime },Q^{\prime }\right)$ can be obtained after the following rotational transformation:(6)$$\begin{array}{ll} \left[\begin{array}{c} I^{\prime }\\ Q^{\prime } \end{array}\right] & =\left[\begin{array}{c} r\cos(\alpha +\beta )\\ r\sin(\alpha +\beta ) \end{array}\right]\\ \; & =\left[\begin{array}{c} r\cos\alpha \cos\beta -r\sin\alpha \sin\beta \\ r\sin\alpha \cos\beta +r\cos\alpha \sin\beta \end{array}\right]\\ \; & =\left[\begin{array}{cc} \cos\alpha & -\sin\alpha \\ \sin\alpha & \cos\alpha \end{array}\right]\left[\begin{array}{c} I\\ Q \end{array}\right], \end{array}$$
where α refers to the angle of rotation, β and *r* refer to the inherent properties of the original signal sample. In this paper, the augmented signals are obtained by rotating the original signal by 0, 90, 180, and 270 degrees in the counterclockwise direction, so a set of original signal samples is augmented into four sets of signal samples.

#### 3.3.2. RandMix

The RandMix data augmentation method works on the signal dataset after augmentation using rotation and is useful as an effective regularization method to improve the generalization performance and robustness of the model. The principle of RandMix data augmentation is shown in Figure 8. For a given signal sample, it is first sliced into *M* signal slices with equal lengths based on its total length. These *M* signal slices are then randomly combined to create an augmented signal sample.

### 3.4. Training Procedure

In summary, based on the above modules, the full training procedure for the proposed AMC method is described in Algorithm 1.
**Algorithm 1** Training procedure of the proposed AMC method.**Require**:   *D*: Training dataset;   *T*: Number of training iterations;   *B*: Number of batches in a training iteration;   *N*: Length of each datum;   *M*: Number of segments for each datum;   θ: Parameters of the lightweight CVResNet;   lr: Learning rate;     **Data augmentation**: 1: $D_{rotate}=\left\{D,D_{\pi /2},D_{pi},D_{3pi/2}\right\}\leftarrow Rotate\;D$;     **Training procedure**: 2: **for** t=0 to T−1 **do**: 3:       **for** b=0 to B−1 **do**: 4:             Sample a batch training dataset (x,y) from $D_{rotate}$.                 **Forward propagation**:                 Evenly split x. 5:             $x\to \left\{x_{0}^{N/M-1},x_{N/M}^{2N/M-1},\cdots ,x_{(M-1)N/M}^{N-1}\right\}=X$;                 Randomly mix X. 6:             $Randmix\;X\to x^{*}$;                 Get the output of lightweight CVResNet. 7:              $\hat{y}=f\left(\theta^{t,b};x^{*}\right);$                 Calculate the loss. 8:              $L=-E\left(y\log\hat{y}\right)$                 **Backward propagation**: 9:              $\theta \leftarrow Adam(\nabla_{\theta },L,lr)$ 10:       **end for** 11:       **end for** 12: **end for** 13: **end for**

## 4. Simulation Results and Analysis

### 4.1. Experiment Environment and Parameters

The simulation was conducted using PyTorch as the backend on a GTX 1080Ti platform. To assess the classification performance and model complexity of the lightweight CVResNet, we used the open-source dataset RadioML 2016.10A [58]. Five modulated scheme data were selected from this dataset, with a modulation scheme model pool of P=BPSK,QPSK,8PSK,QAM16,QAM64 and an SNR of 2 dB. The dataset contained 1000 samples for each modulated type per SNR. For training, validating, and testing, we used 420, 180, and 400 samples, respectively. During the neural network training process, the best model on the validation set was saved as the final classification model. The simulation parameters are shown in detail in Table 1.

The dataset parameters are shown in Table 2.

### 4.2. Simulation Results

#### 4.2.1. Comparative Experiments

This paper compares our AMC method with three comparison methods, i.e., support vector machine (SVM), CVResNet, and lightweight CVResNet. SVM uses support vector machine to classify the original samples, CVResNet uses a non-lightweight network structure to implement classification, and lightweight ResNet uses a lightweight network structure to classify signal samples.

#### 4.2.2. Classification Accuracy: Proposed Method vs. Comparative Methods

The simulation results are shown in Figure 9 and Table 3, and it can be seen that the proposed method has the highest classification accuracy and F1-Score. Compared with SVM and lightweight ResNet, the proposed method can achieve at least 30% improvement; compared with the non-lightweight network CVResNet, it can achieve about 2% improvement.

#### 4.2.3. Confusion Matrix: Proposed Method vs. Comparative Methods

The confusion matrix is an important index used to measure the performance of classification, which can intuitively show the classification of each sample. Figure 10 shows the confusion matrix results of the four methods. It can be seen that the proposed method has the best classification performance.

#### 4.2.4. Ablation Experiments

In this paper, ablation experiments were carried out to verify the effectiveness of the two data augmentation methods. The experimental results are shown in Figure 11 and Table 4. It can be seen that both rotation and RandMix methods can effectively improve the classification performance of the proposed lightweight CVResNet.

#### 4.2.5. Features Visualization

To observe the feature extraction abilities of different methods more intuitively, the features extracted can be visualized by using the t-distributed stochastic neighbor embedding (t-SNE) [59] to reduce the extracted features to two dimensions. The feature map and silhouette coefficient (SC) are shown in Figure 12.

#### 4.2.6. Complexity Analysis

The depthwise separable convolution splits a standard convolution into two convolutions (depth-by-depth, point-by-point), thus reducing the number of parameters and the total computational effort. In this paper, the lightweight model is based on the depthwise separable convolution, so the space complexity and time complexity of the standard convolution and the depthwise separable convolution are analyzed here. The space complexity is measured by the number of parameters in the convolutional layer, and the time complexity is measured by floating point operations (FLOPs). The space complexity and time complexity of the standard convolution are shown as
(7)$$\begin{array}{c} P_{std}∼O\left(K_{h}K_{w}C_{in}C_{out}\right), \end{array}$$
(8)$$\begin{array}{c} T_{std}∼O\left(K_{h}K_{w}C_{in}C_{out}G_{h}G_{w}\right), \end{array}$$
where $P_{std}$ and $T_{std}$, respectively, represent the parameters and FLOPS; $K_{h}$ and $K_{w}$, respectively, represent the two dimensions of the convolution kernel; $C_{in}$ and $C_{out}$, respectively, represent the number of input channels and output channels; $G_{h}$ and $G_{w}$, respectively, represent the two dimensions of the output characteristic graph. Compared to standard convolution, the space complexity and time complexity of the depthwise separable convolution can be expressed as
(9)$$\begin{array}{c} P_{sep}∼O\left(K_{h}K_{w}C_{in}+C_{in}C_{out}\right), \end{array}$$
(10)$$\begin{array}{c} T_{sep}∼O\left(K_{h}K_{w}C_{in}G_{h}G_{w}+C_{in}C_{out}G_{h}G_{w}\right). \end{array}$$

The ratios of the space complexity and time complexity of depthwise separable convolution and standard convolution are as follows.
(11)$$\begin{array}{c} \frac{P_{sep}}{P_{std}}=\frac{K_{h}K_{w}C_{in}+C_{in}C_{out}}{K_{h}K_{w}C_{in}C_{out}}=\frac{1}{C_{out}}+\frac{1}{K_{h}K_{w}}, \end{array}$$
(12)$$\begin{array}{c} \frac{T_{sep}}{T_{std}}=\frac{K_{h}K_{w}C_{in}G_{h}G_{w}\;+\;C_{in}C_{out}G_{h}G_{w}}{K_{h}K_{w}C_{in}C_{out}G_{h}G_{w}}=\frac{1}{C_{out}}+\frac{1}{K_{h}K_{w}}. \end{array}$$

From Equations (Equation 11) and (Equation 12), it can be found that the number of parameters and FLOPS of the depthwise separable convolution is much smaller compared to the standard convolution, so the model can be more lightweight. A detailed comparison of the complexity of the lightweight CVResNet in this paper and the standard CVResNet is shown in Table 5. From the table, we can see that the number of parameters in the lightweight CVResNet has decreased by about 83.34%, and the FLOPs have decreased by about 83.77%, compared to those of the standard CVResNet.

## 5. Conclusions

This article proposes a lightweight and efficient AMC method and validates it using a modulated signal dataset. The experimental results demonstrate that this method has higher classification accuracy, stronger robustness, and better generalization ability compared to three other comparison methods, with the accuracy improved by at least 2%. Additionally, the ablation experiment further confirms the effectiveness of each module in this method, with an accuracy improvement of about 8%. In the complexity analysis, the proposed lightweight AMC method outperforms the non-lightweight method, with advantages in both model complexity and computational complexity. However, the effect of noise on feature extraction is not considered in this paper, so future work will explore various signal-to-noise ratios and other schemes, such as Transformer or LSTM for effective feature extraction from time-series data. Overall, the proposed method has broad applications in various IQ signal-based electromagnetic signal identification tasks, including radiation source identification. Based on complex-valued separable convolution, the network framework can be easily modified or expanded according to the target task requirements, and further performance improvements can be achieved by combining with other data augmentation methods.

## Figures and Tables

**Figure 1 sensors-23-04187-f001:**
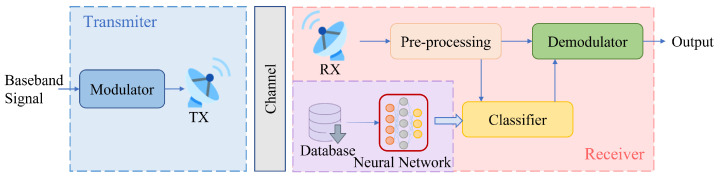
System framework of AMC based on the neural network.

**Figure 2 sensors-23-04187-f002:**
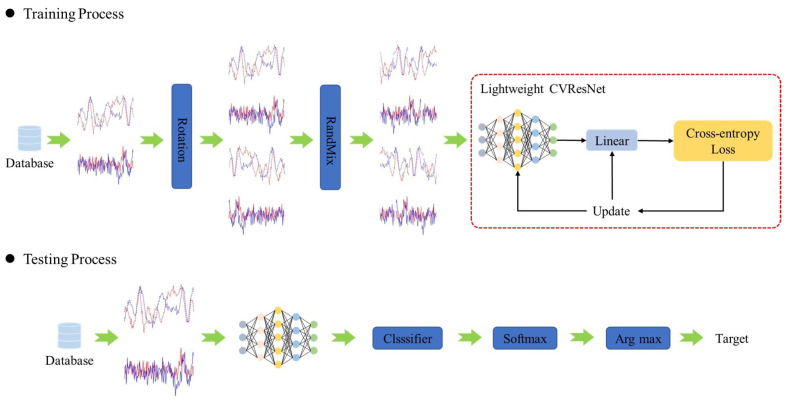
The framework of the lightweight CVResNet.

**Figure 3 sensors-23-04187-f003:**
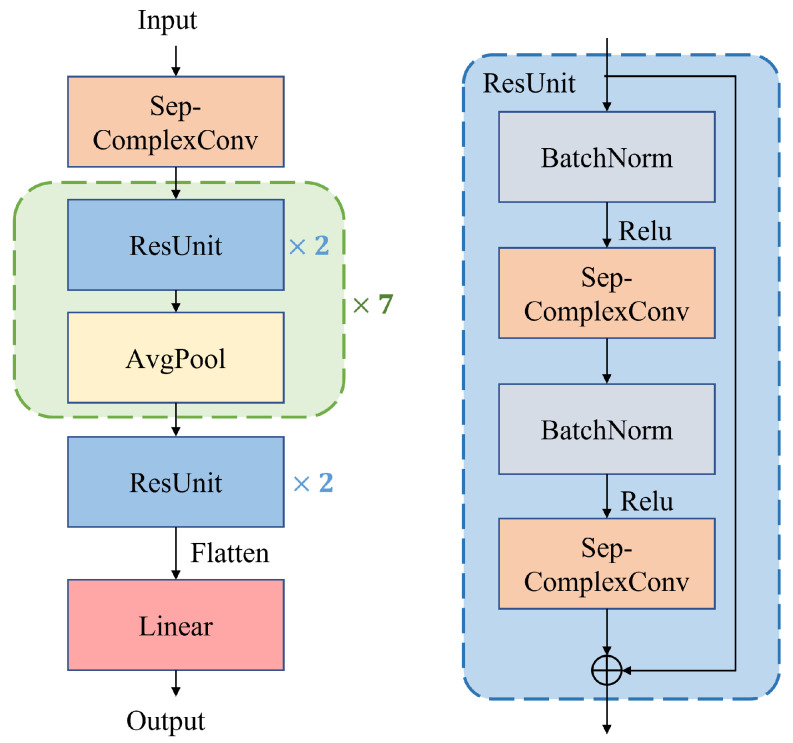
The structure of ResNet.

**Figure 4 sensors-23-04187-f004:**
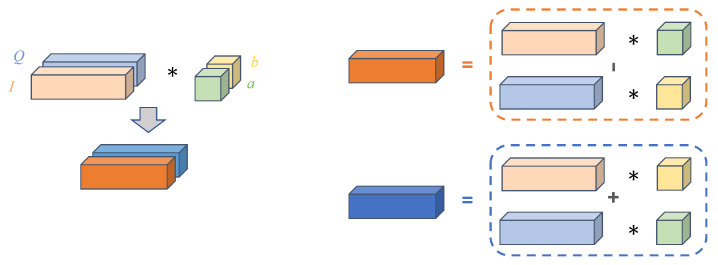
Complex convolution.

**Figure 5 sensors-23-04187-f005:**
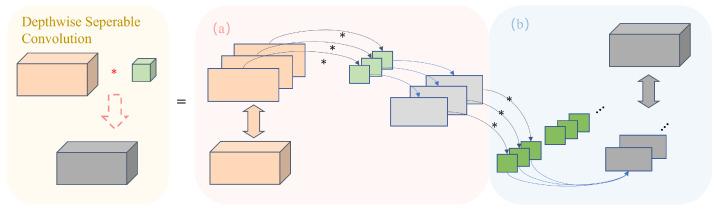
Depthwise separable convolution, where (**a**) depthwise convolution, and (**b**) pointwise convolution.

**Figure 6 sensors-23-04187-f006:**
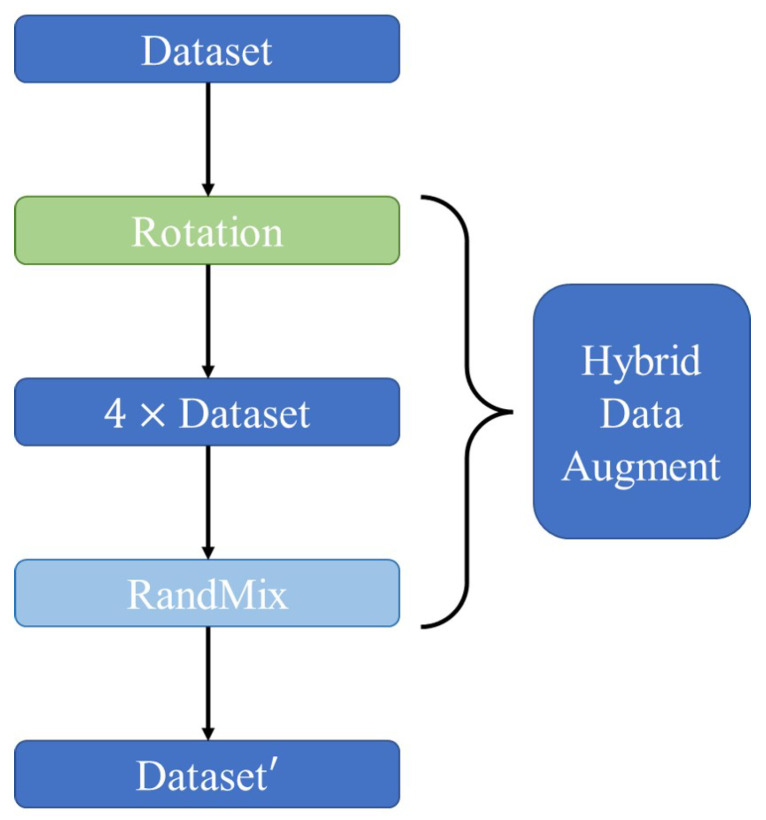
Hybrid data augmentation.

**Figure 7 sensors-23-04187-f007:**
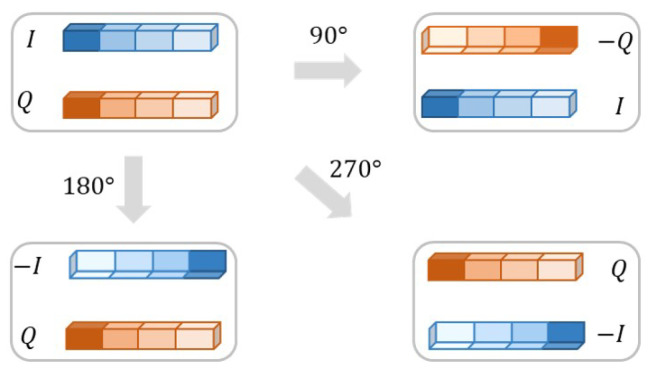
Principle of rotation.

**Figure 8 sensors-23-04187-f008:**
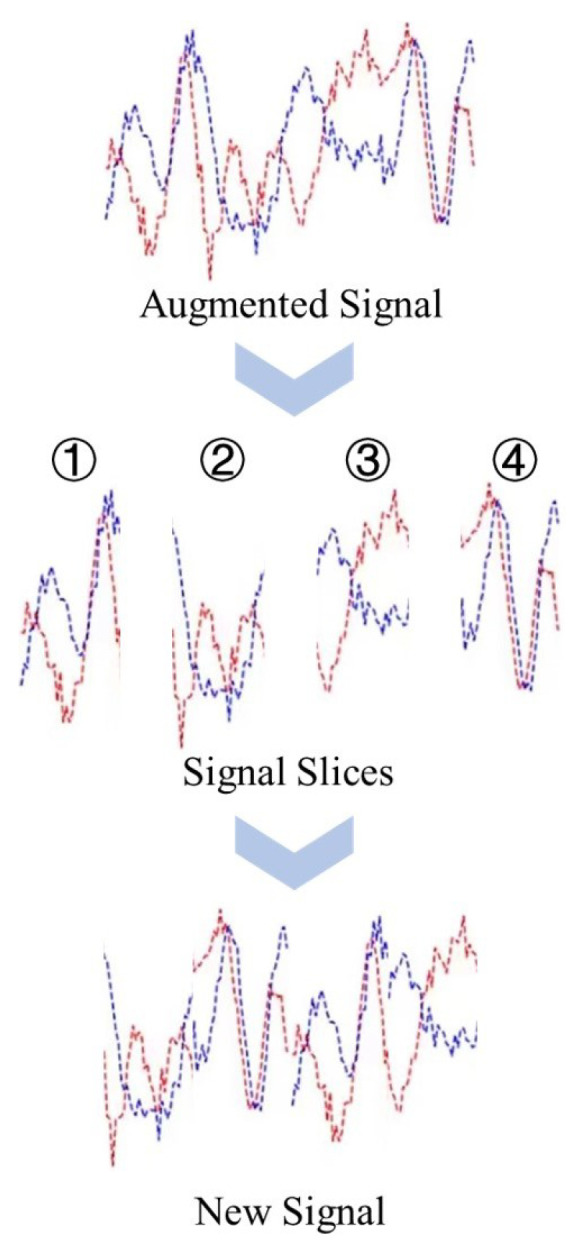
Principle of RandMix.

**Figure 9 sensors-23-04187-f009:**
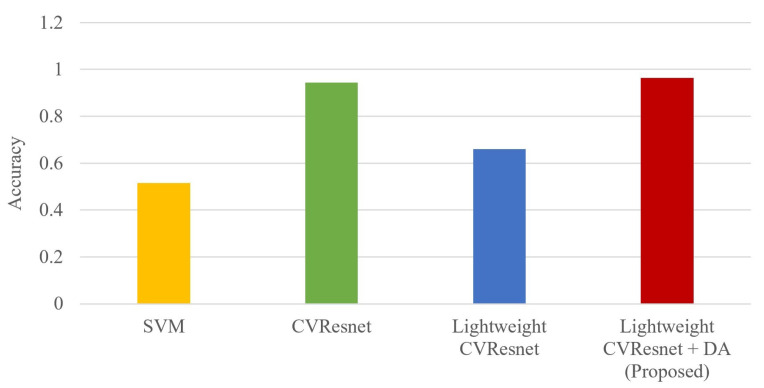
The classification accuracy of the proposed AMC method and comparative AMC methods.

**Figure 10 sensors-23-04187-f010:**
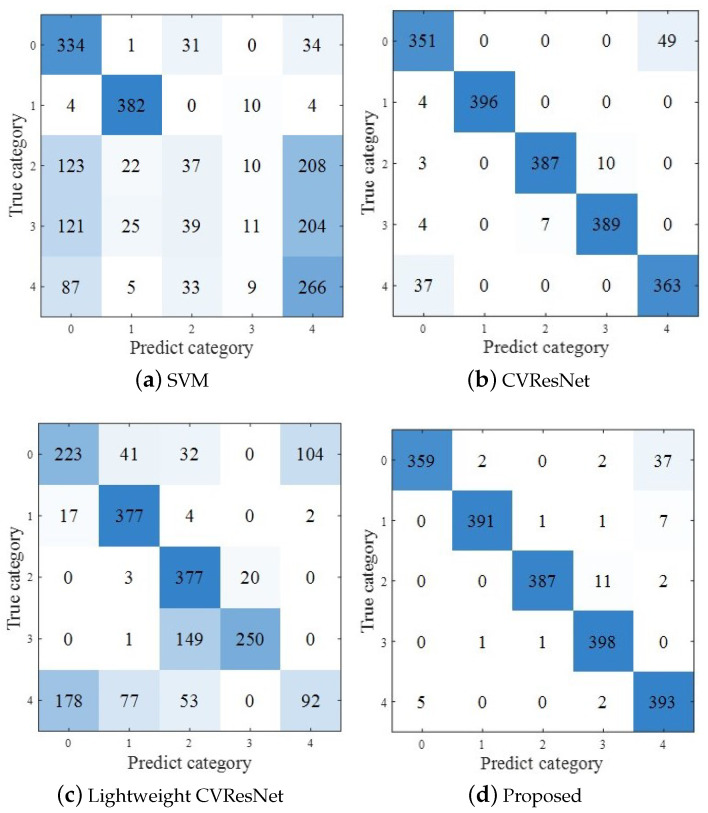
The confusion matrices of the proposed AMC method and comparative AMC methods.

**Figure 11 sensors-23-04187-f011:**
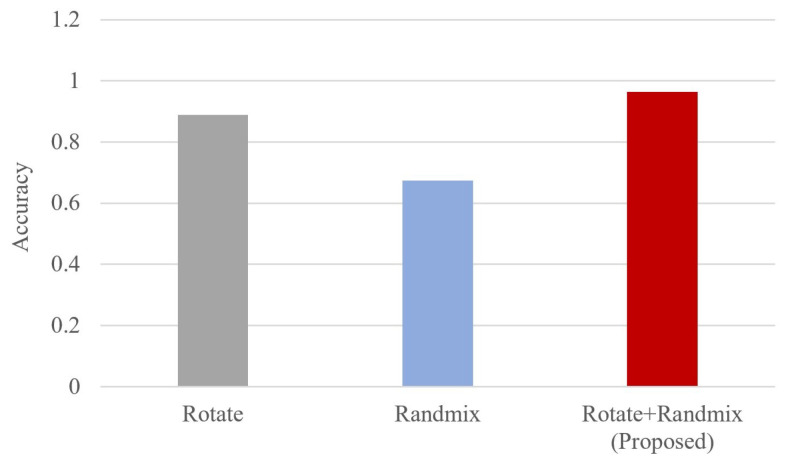
Results of ablation experiments for different methods.

**Figure 12 sensors-23-04187-f012:**
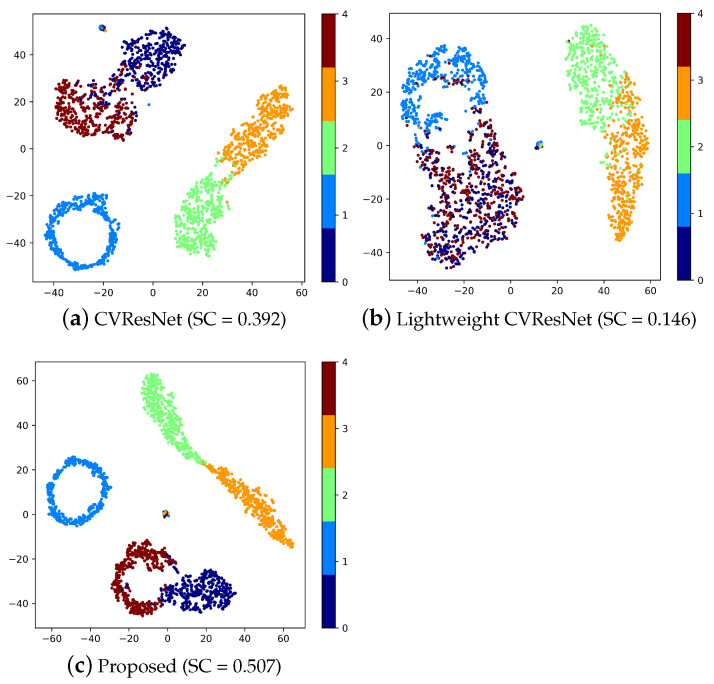
Feature visualization.

**Table 1 sensors-23-04187-t001:** Simulation parameters.

Simulation Parameters	Values
Data dimension	2×128
Number of *D*	3000
Number of $D_{val}$	2000
Optimizer	Adam
Loss	Cross Entropy Loss
Epoch	100
Batch-size	512
Learning rate	0.001

**Table 2 sensors-23-04187-t002:** Dataset parameters.

Simulation Parameters	Contents
Modulation type	BPSK, QPSK, 8PSK, 16QAM, 64QAM
Data format (I/Q)	2×128
Standard deviation of the sampling rate offset	0.01 Hz
Maximum sample rate offset	50 Hz
Carrier frequency offset standard deviation	0.01 Hz
Maximum carrier frequency offset	500 Hz
No. of sine waves in frequency selective fading	8
Sampling rate	200 kHz
Noise	AWGN

**Table 3 sensors-23-04187-t003:** The classification accuracy of the proposed AMC method and comparative AMC methods.

Methods	SVM	CVResNet	Lightweight CVResNet	Proposed
Accuracy	51.50%	94.30%	65.95%	96.40%
F1-Score	0.441	0.943	0.636	0.964

**Table 4 sensors-23-04187-t004:** Results of ablation experiments.

Methods	Rotate	RandMix	Rotate+RandMix (Our Proposed)
Accuracy	88.85%	67.40%	96.40%

**Table 5 sensors-23-04187-t005:** The parameters, FLOPS, model size of CVResNet, and lightweight CVResNet.

Network	Parameters	FLOPs	Size/KB
CVResNet	1,848,965	117,750,272	7376
Lightweight CVResNet	308,117 (83.34%↓)	19,115,008 (83.77%↓)	1424 (80.69%↓)

## Data Availability

The research data can be available via email requirement.

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
