# Peer review of "Automatic Modulation Classification Using Hybrid Data Augmentation and Lightweight Neural Network"

_sensors, 2023, doi:10.3390/s23094187_

Round 1

Reviewer 1 Report

Globally, the manuscript is very well written and organized. However, there are some English errors that must be corrected; please refer to the attached commented PDF file where some of the needed corrections are highlighted. I also recommend a thoughtful reading of the manuscript.

Additionally, to the corrections highlighted in the attached commented PDF file I also recommend the authors to include the percentage of reduction in the number of FLOP operations, as this reduction is very significant. The authors should mention this reduction both at the abstract and at the end of the “4. Simulation Results And Analysis” section.

The title of the "References" section is missing.

Author Response

Thank you very much for your great comments. All of these comments have been addressed in the attached response letter. Hope our response and revised manuscript can be satisfied in the second round of review.

Reviewer 2 Report

The authors proposed a light-weight residual network based on complex-valued operations for automatic modulation classification problem. Additionally, the authors used a hybrid data augmentation method to compensate for the performance drop caused by using the lightweight neural network. The proposed approach was experimentally verified. The results showed that despite using a lightweight network, the proposed approach obtained better classification performance than the other methods. The paper is well-structured. The authors started from the review of related works. On that basis, they defined the research problem and proposed the solution. The proposed method is well presented, and it was experimentally verified and compared with other approaches.

Please provide additional explanations to what extent the proposed approach is specific only to the problem under consideration, and to what extent it could be generalized and applied to other problems.

Please improve the English language when it comes to grammar and style before the publication.

Author Response

(The authors gave the same response as above.)

Reviewer 3 Report

This paper proposes a lightweight neural network-based framework for automatic modulation classification (AMC) to address the challenges of high complexity and parameter count in traditional neural network-based methods. The proposed framework combines complex convolution with residual networks to improve classification performance and uses depthwise separable convolution for lightweight design. A hybrid data augmentation scheme is also proposed to compensate for performance loss caused by lightweight design. In my review, I concentrated on the model design and learning process. 

1. I'm wondering that residual-based network is enough to cover all time intervals. How many samples in your time interval? For the purpose of feature extraction, can you consider other schemes such as Transformer or LSTM that can effectively extract features of time-series data.

2. In your depth-wise separable convolution, I'm not convinced that this separable scheme can sufficiently utilize neighbor information in time domain. Please more specify why this scheme is powerful in spite of not using neighbor time data.

3.  Can you show or verify why rotation data augmentation can improve the generalization? Could be there any other augmentations?

4. Please provide any other metrics such as ROC curve or F1 score to check bias in classification results.

Author Response

(The authors gave the same response as above.)

Reviewer 4 Report

The study is good. There are some points that need to be addressed.

- In the summary, the most important result of the manuscript should be indicated.

- What is the gap in previous studies, which addressed in this study?

- Figure 1 should be redrawn to represent the complete study and make it easier for the reader to understand it.

- In the introduction section, reference should be made to the role of artificial intelligence in smart wireless communications and how to solve manual deficiencies.

- Your results should be compared with previous studies.

- What are the study limitations?

- The authors should mention in future studies.

- The conclusions section is not satisfactory, it should be rewritten in a clearer way.

Author Response

(The authors gave the same response as above.)

Round 2

Reviewer 3 Report

Most of my concerns have been addressed. But I have a few comments to reach the publication level.

1. The samples are relevant to the sampling ratio. How many discrete samples in the time domain? If I were you, I would suggest input data size instead.

2. It would be better to show the ablation studies applying other augmentations such as flip or crop.

3. There are typo in Table 2. (The scale of F1 score seems to be incorrect.) 

Author Response

Dear Reviewer:

Many thanks for providing great comments to improve this paper. All of your comments are replied to below and the revised manuscript has been attached.

Comments 1. The samples are relevant to the sampling ratio. How many discrete samples in the time domain? If I were you, I would suggest input data size instead.

Response:  Thank you very much for your great comment. The used dataset is originated from the open-source dataset RadioML 2016.10A [45]. The sampling ratio is 200kHz. All of the dataset parameters are set in Table 2. 

Comment 2. It would be better to show the ablation studies applying other augmentations such as flip or crop.

Response: Thank you very much for the great comments. This paper shows the typical case of an ablation experiment (Rotate in Figure 11) which is one of the cases of flip. Our proposed method using (Rotate+Randmix in Figure 11) is better than the method using Rotate. Hope our explanation is suitable. Many thanks for the kind suggestion.

Comment 3. There are typos in Table 2. (The scale of the F1 score seems to be incorrect.) 

Response: Many thank for the correction. We have corrected the mistake in the revised manuscript in Table 3 (It is Table 3 in the revised manuscript).  

Reviewer 4 Report

The authors responded to the comments and the manuscript was improved.

accept in present form

Author Response

Dear Reviewer,

Thank you very much for providing comments to improve this paper. We shall continue working hard on this topic. 

Yours sincerely

The authors